# Effect of combined acetylsalicylic acid and statins treatment on intracranial aneurysm rupture

Mikel Terceño[1,2], Sebastian Remollo[2], Yolanda Silva[1]*, Saima Bashir[1], Mariano Werner[2], Víctor A. Vera-Monge[1], Joaquín Serena[1], Carlos Castaño[1]

1 Stroke Unit, Department of Neurology Girona Biomedical Research Institute(IDIBGI), Dr Josep Trueta University Hospital, Girona, Spain, 2 Interventional Neuroradiology Unit, Department of Neurosciences, Germans Trias i Pujol University Hospital, Badalona, Spain

* ysilva.girona.ics@gencat.cat

## Abstract

### Background

Acetylsalicylic acid (ASA) and statins have been identified as potentially reducing the risk of intracranial aneurysms (IA) rupture. We aim to determine the effect of this drugs on the risk of rupture of IA.

### Patients and methods

We performed a retrospective cohort study from a prospective database of patients with IA treated in our institution between January 2013 and December 2018. Demographics, previous oral treatments, presence of multiple aneurysms, size of aneurysm, lobulation, location and morphology of the aneurysms were recorded. Patients were dichotomized as ruptured and unruptured IA.

### Results

A total of 408 IA were treated, of which 283 (68.6%) were in women. The median age was 53, 194 (47.5%) were ruptured IA. 38 patients (9.3%) were receiving ASA and 84 (20.6%) were receiving statins at the moment of the IA diagnosis. In the multivariable regression analysis, ASA plus statin use and multiple aneurysms were independently associated with unruptured IA (OR 5.01, 95% CI, 1.37–18.33, P = 0.015 and OR 2.72, 95% CI 1.68–4.27, P<0.001, respectively). Whereas, lobulated wall aneurysm and PComA/AComA location were inversely and independently associated with unruptured IA condition (OR 0.34, 95% CI 0.21–0.55, P<0.001 and OR 0.37, 95% CI 0.23–0.60, P<0.001, respectively). However, ASA and statins in monotherapy were not independently associated with unruptured IA condition.

### Conclusions

In our study population ASA plus statins treatment is independently associated with unruptured IA. Larger and prospective studies are required to explore this potential protective effect against IA rupture.

**Data Availability Statement:** All rellevant data are within the paper and its Supporting Information file.

**Funding:** The authors received no specific funding for this work.

**Competing interests:** The authors have declared that no competing interests exist.

# Background

Approximately 3% of the population have an unruptured intracranial aneurysm (IA). Factors such as posterior location, lobulated morphology, inflammation of the aneurysm wall, large dome aneurysm size, previous subarachnoid hemorrhage (SAH) and smoking have been identified as risk factors for the prediction of IA rupture [1–3].

In the last few years, the study of the effect of statins and especially of acetylsalicylic acid (ASA) on the risk of IA rupture have shown promising results in preventing aneurysm rupture and growth. The pleiotropic effect consisting in an endothelium protection by increasing nitric oxide (NO) and by inhibiting cytokines and matrix metalloproteinases (MMPs), and the mobilization of endothelial progenitor cells (EPC) have been associated to statins, resulting in an inhibited endothelial injury and media thinning of the aneurysm. On the other hand, some studies have suggested that unselective inhibition of cyclooxygenase 2 (COX-2) and prostaglandin E2 synthase-1 with ASA can result in a reduction of wall inflammation that leads to a drop in IA rupture rates. These mechanisms associated with ASA and statins, have been hypothesized as the potential causes to explain the reduction of IA rupture and growth in some studies [4–10].

Currently, there are some ongoing clinical trials trying to confirm these effects and their implication on humans, by studying the potential benefit of statins and ASA in unruptured IA, as single therapy [11,12].

In this study we aim to compare the effect of both treatments and explore the impact of a combined treatment (ASA plus statins) on IA rupture.

# Patients and methods

We performed a retrospective cohort study from a prospective database of patients with IA treated in our institution between January 2013 and December 2018.

From 428 consecutive patients with 473 IA admitted, a total of 368 patients with 408 IA were treated and included in the analysis. This retrospective consecutively recorded study was approved by the local research ethics committee (Comitè d'Ètica de la Investigació amb Medicaments at Germans Trias i Pujol University Hospital, Badalona, Spain) in October 2018.

All patients or legal representatives signed the corresponding informed consent.

The inclusion criteria were patients >18 years old, with IA treated endovascularly in our institution, known drugs treatment at diagnosis and follow-up completed at 12 months.

Demographic, clinical characteristics and previous medical treatments were collected. Variables related to the aneurysm, such as dome diameter, dome/neck ratio, artery location, presence of lobulations and coexistence of multiple aneurysms, were evaluated by cerebral angiography.

Patients included in the analysis as ASA or statins users, were taking these drugs daily at least one year before IA diagnosis. In patients with ruptured aneurysms, medication history was obtained through a patient interview when possible or based on family interviews and previous medical reports.

We dichotomized patients in two groups: ruptured and unruptured aneurysms.

A ruptured aneurysm was considered if subarachnoid haemorrhage (SAH) on CT study was present and it was classified in four different groups by using the Fisher scale.

Follow-up and mortality for all cases were recorded at 12 months.

The goal of our study was to determine the role of combining ASA plus statins as potentially protective factor of IA rupture and their comparison with any monotherapy.

## Statistical analysis

The baseline characteristics of the patients included in the study were compared using the Student t test or the Mann–Whitney U test for continuous variables and the $\chi 2$ test for categorical

variables. Univariate analysis was performed to study variables associated with IA rupture. Continuous variables were reported as mean±SD or median (inter- quartile range), as appropriate. Categorical variables were reported as proportions.

Logistic regression was used to assess the association of risk factors with the IA rupture. Variables with P values less than 0.10 were entered in the multivariate logistic models with a forward stepwise procedure. Data were analysed with SPSS version 21 software and all tests were performed with a 5% significance level.

## Results

368 patients with 408 IA were included in the study. The median age was 53 years, and 68.6% were female. 214 of the IAs (52.5%) were unruptured at the moment of diagnosis. The most frequent ASA dose was 100 mg per day (94.7%) and Simvastatin was the statin most frequently used (52.4%). Demographics and vascular risk factors were similar in both groups with no statistically significant differences (Table 1).

Patients from the unruptured IA group more frequently had multiple IAs (56.8% vs. 35.8%, P<0.001) at baseline angiography. In addition, these patients were more frequently taking statins (26.2% vs 14.4%, P = 0.005) and ASA (12.6% vs 5.7%, P = 0.017). The combination of taking ASA plus statins at diagnosis was significantly higher in these patients (12.1% vs 3.1%, P = 0.001).

The differences between treatment at baseline and aneurysm rupture are represented in Fig 1.

The group of patients with unruptured IA had lower aneurismal dome diameters (7.1 mm vs 6.7 mm, P<0.001) and these aneurysms less frequently had a saccular morphology (87.7% vs 96.9%, P = 0.001) and lower presence of lobulated walls (47.2% vs 73.2%, P<0.001).

**Table 1. Univariate analysis according to aneurysm state at diagnosis.**

| Characteristics | Ruptured IA n = 194 | Unruptured IA n = 214 | P value |
|---|---|---|---|
| Age, years | 54 [46–66] | 53 [46–66] | 0.549 |
| Sex, female | 128 (66%) | 152 (71%) | 0.287 |
| HBP | 81 (41.8%) | 96 (44.9%) | 0.549 |
| Alcoholism | 13 (6.7%) | 16 (7.5%) | 0.761 |
| Smoking | 77 (39.7%) | 98 (45.8%) | 0.119 |
| Diabetes Mellitus | 10 (5.2%) | 13 (6.1%) | 0.830 |
| Dyslipidemia | 56 (28.9%) | 65 (30.4%) | 0.746 |
| Ischemic cardiopathy | 6 (3.1%) | 15 (7%) | 0.057 |
| Statins treatment | 28 (14.4%) | 56 (26.2%) | **0.005** |
| ASA treatment | 11 (5.7%) | 27 (12.6%) | **0.017** |
| ASA plus statin treatment | 6 (3.1%) | 26 (12.1%) | **0.001** |
| Multiple IAs | 69 (35.8%) | 121 (56.8%) | **<0.001** |
| Dome diameter, mm | 6.7±3.51 | 7.1±5.91 | **<0.001** |
| Dome-neck ratio | 1.8 [1.4–2.3] | 1.5 [1.1–2.0] | 0.215 |
| Saccular morphology | 188 (96.9%) | 186 (87.7%) | **0.001** |
| Lobulated wall aneurysm | 142 (73.2%) | 101 (47.2%) | **<0.001** |
| PComA/AComA location | 103 (53.1%) | 50 (23.4%) | **<0.001** |
| Posterior circulation location | 22 (11.3%) | 24 (11.2%) | 0.472 |
| Mortality at 12 months | 38 (19.6%) | 0 (0%) | **<0.001** |

Continuous variables are expressed as mean ± SD or median [quartiles] as appropriate or n (%).

Abbreviations: HBP, high blood pressure; IAs, intracranial aneurysms; ASA, acetyl-salicylic acid; PComA, posterior communicating artery; AComA, anterior communicating artery.

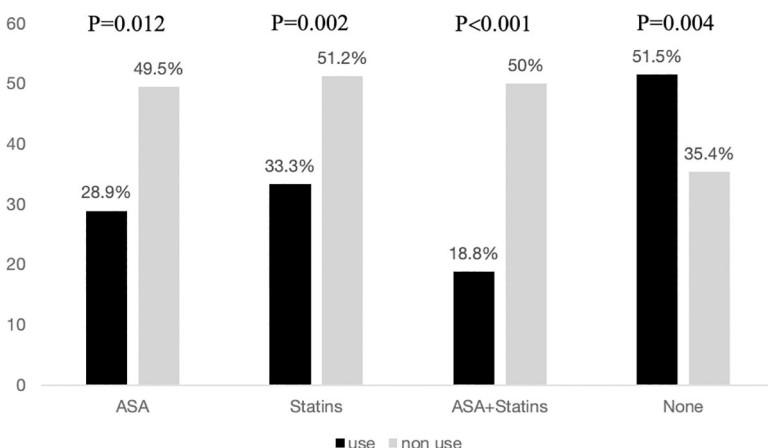

**Fig 1. Percentages of IA rupture according to treatment at aneurysm diagnosis.**

A total of 32 (7.8%) patients were taking the combined treatment (ASA plus statins) at the moment of diagnosis. This treatment was more frequent in the group of unruptured IA (12.1% vs 3.1%, P = 0.001). No deaths were registered at 12 months follow-up in the group of unruptured IA, compared to a mortality of 19.6% in the ruptured IA group (p<0.001).

After adjusting for multiple variables in the regression model, combined treatment was identified as an independent predictor of unruptured IA (adjusted odds ratio, 5.01, 95% CI, 1.37–18.33, P = 0.015), together with the presence of multiple IAs (adjusted odds ratio, 2.72, 95% CI, 1.68–4.27, P<0.001), whereas the use of ASA or statins in monotherapy were not independent predictors of unruptured IA (adjusted odds ratio, 1.22, 95% CI, 0.42–3.53, P = 0.715 and adjusted odds ratio, 1.65, 95% CI, 0.83–3.31, P = 0.155, respectively) (Table 2).

## Discussion

In this study we show that the combined treatment has a higher protective effect against rupture over any single therapy in patients with IA. ASA and statins have been reported as potential therapies in preventing aneurysm rupture and growth. However, this is the first study that evaluate both treatments as a combined therapy, based on the synergist effect of ASA and statins on the aneurysm endothelium.

**Table 2. Multivariate regression model for unruptured intracranial aneurysm at diagnosis.**

| Variable | Adjusted OR | 95% CI | P value |
|---|---|---|---|
| Ischemic cardiopathy | 1.21 | 0.31–4.70 | 0.781 |
| ASA treatment | 1.22 | 0.42–3.53 | 0.715 |
| Statins treatment | 1.65 | 0.83–3.31 | 0.155 |
| ASA plus statin treatment | 5.01 | 1.37–18.33 | **0.015** |
| Multiple IAs | 2.72 | 1.68–4.27 | **<0.001** |
| Dome diameter | 0.97 | 0.92–1.03 | 0.339 |
| Saccular morphology | 0.42 | 0.14–1.24 | 0.115 |
| Lobulated wall aneurysm | 0.34 | 0.21–0.55 | **<0.001** |
| PComA/AComA location | 0.37 | 0.23–0.60 | **<0.001** |

*Adjusted by confounding factors, with a P < 0.10 in the univariate analysis.

Abbreviations: ASA, acetyl-salicylic acid; IAs, intracranial aneurysms PComA, posterior communicating artery; AComA, anterior communicating artery.

Inflammation of the aneurysm wall has been identified with aneurysm growth and rupture. This process is started by the action of several mediators and leads to a degradation of the extracellular matrix by matrix metalloproteinases (MMPs) and the destruction of smooth muscle cells (SMC), leading to wall thinning and, consequently, a risk of aneurysm rupture. Macrophage infiltration into the aneurysm wall is considered to be a key factor in IA formation and growth [13–19].

## Evidence of ASA

The antiplatelet and anti-inflammatory mechanisms of ASA are widely known. ASA blocks the action of Cyclooxygenase-2 (COX-2), which converts arachidonic acid from prostaglandins, resulting in Ca2+ influx into the endothelial cell in the aneurysm wall. Thus, COX-2, has been considered in some studies as a potential target to decrease the inflammatory phenomena that takes place in the aneurysm [19,20].

ASA inhibits the migration of macrophages into the aneurysm wall and has been shown to suppress MMP-2 and MMP-9 expression [21–24].

The clinical benefit of ASA in preventing risk of unruptured IA has been studied since 2011 when the ISUIA (International Study of Unruptured Intracranial Aneurysms) investigators determined that patients who used ASA at least three times weekly had significantly lower rates of SAH compared to patients who never used ASA [7].

Zanaty et al., studied a population with 229 unruptured IAs, finding that aneurysm growth was observed in 10.5% of them during follow-up. ASA use was identified as the only variable independently associated with non-aneurysm growth [10].

In line with these results, Gross et al., in a study of 747 consecutive patients with unruptured and ruptured IA, found a statistically significant association between ASA use and unruptured IA, with no differences in outcome at one year [25]. Hostettler et al. obtained similar results, finding an inverse association between ASA use and rupture status [8].

The largest case-control was published by Can et al. in 2018. They studied 4,708 patients with 6,411 ruptured and unruptured IA. In the multivariable analysis, ASA use was associated with lower rates of SAH. On the other hand, ASA use was identified as a significant factor associated to rebleeding, however, its impact on mortality was not described [9].

The preventive effect of the ASA effect on IA risk rupture has been studied in prospective animal models [14,26,27].

Despite these promising results, there is no consistent data nor international guidelines to recommend ASA use to patients with a diagnosis of unruptured IA, as once the IA is ruptured, patients with SAH and undergoing ASA could have poorer outcomes [27], although, this association has not been confirmed by some other studies [28].

## Evidence of statins

The effect of statins on unruptured IA has also been studied although these investigations have mainly focused on the potential benefit in preventing cerebral vasospasm secondary to IA rupture [29–31].

The pleiotropic effect of statins is found to protect the endothelium by increasing nitric oxide (NO) and by inhibiting proinflammatory cytokines, MMPs and isoform Nitric Oxid Synthase (iNOS) in macrophages and smooth muscle cells (SMC) [32,33].

In animal models, some histological changes have been obtained, such as an increase in the media thickness and a decrease in the size of aneurysms and in the formation of new ones [34–37].

Can et al. also published their investigations with statins. From the same registry of 6,411 IA, lipid-lowering agent use was significantly inversely associated with rupture status, together with female sex, tobacco use and alcohol use, among others [4].

Conversely, Marbacher et al, conducted a case-control study with 1,200 patients. However, no protecting effect was observed regards IA formation in the group of statin users [38].

Finally, Bekelis et al. completed a study of 28,131 patients with unruptured IA. They concluded that statins use was not associated with a difference in SAH risk although mortality in this group was lower [39].

Due to these previous evidence, we aimed to investigate the effect of the combined treatment (ASA plus statins) in patients with IAs and explore the potential benefit of the combined treatment as a protecting factor against aneurysm rupture.

The potential benefit of the combined treatment, hypothetically, could be due to the different mechanisms in protecting the endothelium: the pleiotropic effect by statins and inhibiting COX-2 by ASA, among others [21,24,32,33].

In our series, the group of patients with unruptured IA at diagnosis presented higher rates of statin and ASA in monotherapy than the ruptured IA group, with no significant differences in vascular risk factors or demographics. In addition, the combined therapy was also higher in the unruptured IA group (12.1% vs 3.1%, P = 0.001). Ischemic cardiopathy was more frequently in the unruptured IA group and, although no statistically significant differences were observed, it could explain a higher consumption of ASA and statins in this group.

Classical variables related to aneurysm, such as the presence of lobulation on the aneurysm wall and the PComA/AComA locations were identified as predictive risk factors for IA rupture, in accordance with previous literature [1–3]. However, smoking and HBP were not associated with IA rupture, in contrast to some previous studies, although, for some investigators, HBP could be associated with IA formation but not with IA rupture [13]. Moreover, a recent study identify uncontrolled hypertension but not controlled hypertension, as an independent predictor of aneurysm growth [40].

Also, in our series dome size was not associated with IA rupture, actually, dome diameter was smaller in the ruptured IA group (6.7 vs 7.1 mm, P<0.001), maybe reflecting a higher importance of other anatomical variables like aneurysm/parent artery ratio and flow angle, that have been described recently and associated with IA rupture [41,42].

The presence of multiple aneurysms was significantly more frequent in the unruptured IA group. Patients with multiple aneurysms had significantly less PComA/AComA aneurysm locations (30.5% vs 43.5%, p = 0.005), and these patients were less frequent ASA users at the moment of the diagnosis (7.4% vs 11.1%, p = 0.13), so we consider that these two variables could influence in the association observed between multiple aneurysm and unruptured IA condition.

When comparing those variables statistically associated with unruptured IA, in the logistic regression model, the combined treatment remains superior to ASA and statins treatments alone revealing a potential synergistic benefit of the two treatments.

An important concern before recommending a combined treatment, is the potential risk of greater neurological and radiological deterioration in patients undergoing ASA, due to the potential impact on haemostasis. This effect has been studied previously with inconclusive results. Tai et al. published a series of 160 patients with ruptured IA, observing that previous SAH, alcohol use and ASA use were independently associated with higher Hunt and Hess scores. On the other hand, smoking, inflow angle <90 degrees but not ASA use, were independently associated with higher Fisher grades [28]. Bruder et al. studied the effect of ASA in 1,422 patients treated for aneurismal SAH. ASA use patients had aneurismal rebleeding more often but with no differences in clinical outcomes [43].

## Limitations and strengths

The small sample size in the group of combined treatment and the fact that this is a single-center and cross-sectional study, are the main limitations of our study. Patients included in the study were all of them treated endovascularly, so our conclusions should be carefully interpreted in in case of patients who are not good candidates for endovascular treatment.

The main strengths are that this is the first study to explore the potential benefit of a combined treatment and compares the protective effect of ASA and statins, also this is a high detailed registry with 12 months follow-up in all patients. If our results are replicated, the combination of ASA plus statins could be beneficial in patients with unruptured IA, becoming a new first-line strategy in preventing IA rupture.

## Conclusions

In our study population ASA plus statins treatment is independently associated with unruptured IA at diagnosis, and this combination is superior to any monotherapy.

Larger prospective studies and controlled clinical trials are necessary to demonstrate the potential benefit of the combined treatment, as well as if some particular statin and dose, could be significantly superior in preventing IA rupture.

## Supporting information

**S1 File.**
(XLSX)

## Acknowledgments

We would like to thank all the patients and their relatives for their participation in the project.

## Author Contributions

**Conceptualization:** Mikel Terceño, Sebastian Remollo, Yolanda Silva, Saima Bashir, Mariano Werner, Víctor A. Vera-Monge, Joaquín Serena, Carlos Castaño.

**Data curation:** Mikel Terceño, Yolanda Silva.

**Formal analysis:** Mikel Terceño, Yolanda Silva.

**Investigation:** Mikel Terceño.

**Methodology:** Mikel Terceño, Sebastian Remollo, Yolanda Silva, Saima Bashir, Mariano Werner, Víctor A. Vera-Monge, Joaquín Serena, Carlos Castaño.

**Resources:** Mikel Terceño.

**Software:** Mikel Terceño.

**Supervision:** Mikel Terceño, Yolanda Silva.

**Writing – original draft:** Mikel Terceño, Yolanda Silva.

**Writing – review & editing:** Mikel Terceño, Sebastian Remollo, Yolanda Silva, Saima Bashir, Mariano Werner, Víctor A. Vera-Monge, Joaquín Serena, Carlos Castaño.

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
