## [Decision Letter · Decision Letter 0]

13 Jan 2021

PONE-D-20-30637

EFFECT OF COMBINED STATINS AND ACETYL SALICYLICACID TREATMENT ON INTRACRANIAL ANEURYSM RUPTURE

PLOS ONE

Dear Dr. Silva,

Thank you for submitting your manuscript to PLOS ONE. After careful consideration, we feel that it has merit but does not fully meet PLOS ONE’s publication criteria as it currently stands. Therefore, we invite you to submit a revised version of the manuscript that addresses the points raised during the review process.

We look forward to receiving your revised manuscript.

Kind regards,

Massimiliano Toscano

Academic Editor

PLOS ONE

Additional Editor Comments:

The main theme is interesting as the paper introduces a novel point of view in the scenery of unruptured intracranial aneurysms. Anyway, there are several major concerns, so that substantial reviews are needed. In the discussion section, findings were not really discussed according to the literature. Moreover, the introduction and methods are rather inaccurate.

Please reply accurately to the questions raised and, most of all, review the whole paper as well as the limitation section taking into account the reviewers’ comments (e.g. no follow-up period, no untreated patients, the small size of patients population and so on).

2. Thank you for including your ethics statement:  "This retrospective consecutively recorded study was approved by the local research ethics committee.".   

"No."

"No"

Reviewers' comments:

Reviewer's Responses to Questions

**Comments to the Author**

1. Is the manuscript technically sound, and do the data support the conclusions?

Reviewer #1: Partly

Reviewer #2: Yes

Reviewer #3: Yes

2. Has the statistical analysis been performed appropriately and rigorously? 

Reviewer #1: Yes

Reviewer #2: Yes

Reviewer #3: Yes

3. Have the authors made all data underlying the findings in their manuscript fully available?

Reviewer #1: Yes

Reviewer #2: Yes

Reviewer #3: Yes

4. Is the manuscript presented in an intelligible fashion and written in standard English?

Reviewer #1: Yes

Reviewer #2: Yes

Reviewer #3: Yes

5. Review Comments to the Author

Reviewer #1: This paper from Terçeno et al. aimed to investigate the predictive role of the treatment with aspirin and statins in reducing the risk of rupture of intracranial aneurysms (IA).

This is a cohort study, retrospectively analysed, of 368 patients with 408 IA treated for 5 years, of which 52.5% unruptured at the time of diagnosis. Patients with unruptured IA more frequently had multiple aneurysms, more frequently taking ASA and statins. 9.3% of patients were receiving ASA and 20.3% were receiving statins at the time of diagnosis. In multivariable analysis, the association therapy of ASA and statins were independently associated with unruptured IA (OR 5.01, 95% CI 1.37-18.33). Other characteristics of the IA were correlated with unruptured IA (lobulated wall aneurysm, PComA/AComA location and multiple aneurysms) and were founded to be independently associated with unruptured IA. Mortality was around 20% at one-year follow-up in ruptured IA. Authors concluded for a potential protective role of the association of statins and ASA treatment in IA rupture.

Although the work is interesting and substantially in line with recent literature findings, the paper is inaccurate in some parts and should be implemented in its formal presentation.

Abstract- In conclusions section, authors stated that association treatment of statins plus ASA is more frequent in unruptured IA but this datum is not presented in the results section of the same abstract. It might be better to state that statins plus ASA association is independently associated with unruptured IA.

Main text

Background section should be more developed.

Methods.

• How authors defined mortality? Did they consider mortality for all causes? Or related to SAH?

• In the statistical analysis section, authors declared that risk factors were associated with IA rupture and Fisher 4 in ruptured IA, but the results of the association between IA rupture and Fisher 4 were not presented in text nor tables in the results section.

• Authors stated that a backward and forward stepwise procedure was used for multivariable logistic regression. Which approach was finally used in table 2?

Results.

• Authors stated that mortality is 19.6% in the ruptured IA at 12 months in the text and at 3 months in table 1. This discrepancy should be solved.

• Mean ASA dose should be reported.

Table 2. Statins treatment showed a significant different distribution in ruptured and unruptured IA (14.4 vs 26.2%, p=0.005) in univariate analysis (table 1). Why statins treatment alone was not inserted as independent variable in multivariable regression models?

Discussion section should be more fluent and less structured to be more enjoyable for readers.

Disappointingly, findings were not really discussed according to literature. Here, some examples:

• Why ASA alone did not result protective in this population, as expected from literature data? Is it in relation with sample size? Is it related to selection bias? Why patients were taking ASA at the time of diagnosis? All these aspects should be discussed. Moreover, ASA dose should be reported and discussed since an inverse dose-response relationship with SAH risk has been reported (Reference 9).

• The same should be done for statins alone, on which far less evidence is available.

• Another interesting point is that dome diameter of unruptured IA was significantly larger than ruptured IA, which is not expected. Moreover, the balanced distribution of high blood pressure between the two groups is not expected since well-controlled blood pressure are associated with a low risk of unruptured IA growth (Weng GC. Aspirin and Growth of Small Unruptured Intracranial Aneurysm: Results of a Prospective Cohort Study. Stroke. 2020 Oct;51(10):3045-3054. doi: 10.1161/STROKEAHA.120.029967. Epub 2020 Sep 3).

• Another main limitation that should be added is the cross- sectional nature of the study.

Reviewer #2: The authors presents a single institution study of cerebral aneurysms and report that a combination of ASA and a statin is associated with a lower incidence of aneurysm rupture. Given the general interest in ASA and statins for preventative care of cerebral aneurysms, this manuscript is worthy of publication if the following comments are addressed.

1) A major limitation to this study is that the patient population is limited to patients who underwent treatment of their aneurysms. This data can therefore not be generalized to all cerebral aneurysm patients. Please discuss this in the "limitations and strengths" section.

2) Please expand your final Conclusion to provide a brief summary of your data (ie, that the combination of ASA and a statin was more frequently seen in unruptured aneurysms).

3) Multiple aneurysms has been associated with a higher risk of aneurysm rupture, and yet you saw the opposite in your data. Do you have any thoughts on this?

4) Including Figure 1 places an emphasis on mortality data, although you found no statistically significant differences in mortality based on ASA and statin use. I would suggest either removing this figure and reporting this data in a table or text or adding error bars to the graph to make it more clear that there are no statistical differences.

Reviewer #3: The authors present a well-written manuscript on the effects of statins and ASA treatment on the rupture of intracranial aneurysms. This is a retrospective review that looks at risk factors and treatment with either statin, ASA or a combination in two groups of aneurysms- ruptured and unruptured. Based on rupture status alone, the authors observe a protective effect of combination treatment. This is a modestly sized study that is retrospective in nature. Additionally there is no follow-up period for patients. I think looking at the data in terms of predictors (ie treatment with statin and/or ASA) does provide some valuable information, and provides some of the seed data needed to perform an RCT of these two medications for treatment. However, I think we still don't truly know from this study if the medications are protective because there is no way of know how long any of the patients harbored an aneurysm, and there is no data available to perform survival analysis with a Cox model. Nonetheless, I think this manuscript serves as an excellent review of the potential effects of ASA and statin on the biology of intracranial aneurysms, and the data helps provide more impetus to move towards an RCT. I would ask that the authors include some of my comments in the limitations section, and flesh out the limitations of the study further in the discussion. I think the conclusions are appropriate for the study design and data.

6. PLOS authors have the option to publish the peer review history of their article (what does this mean?). If published, this will include your full peer review and any attached files.

Reviewer #1: No

Reviewer #2: No

Reviewer #3: **Yes: **Joshua W Osbun

---

## [Author Response · Author response to Decision Letter 0]

30 Jan 2021

PONE-D-20-30637

EFFECT OF COMBINED ACETYLSALICYLIC ACID AND STATINS TREATMENT ON INTRACRANIAL ANEURYSM RUPTURE.

Editor Comments:

Q. The main theme is interesting as the paper introduces a novel point of view in the scenery of unruptured intracranial aneurysms. Anyway, there are several major concerns, so that substantial reviews are needed. In the discussion section, findings were not really discussed according to the literature. Moreover, the introduction and methods are rather inaccurate.

Please reply accurately to the questions raised and, most of all, review the whole paper as well as the limitation section taking into account the reviewers’ comments (e.g. no follow-up period, no untreated patients, the small size of patients population and so on).

A. In this revised manuscript version, we have modified the Methods section, but also the Introduction and the Discussion sections, in order to better respond the comments and to clarify the information. Thanks.

Reviewers' comments:

Reviewer's Responses to Questions

Comments to the Author

Reviewer #1: This paper from Terçeno et al. aimed to investigate the predictive role of the treatment with aspirin and statins in reducing the risk of rupture of intracranial aneurysms (IA).

This is a cohort study, retrospectively analysed, of 368 patients with 408 IA treated for 5 years, of which 52.5% unruptured at the time of diagnosis. Patients with unruptured IA more frequently had multiple aneurysms, more frequently taking ASA and statins. 9.3% of patients were receiving ASA and 20.3% were receiving statins at the time of diagnosis. In multivariable analysis, the association therapy of ASA and statins were independently associated with unruptured IA (OR 5.01, 95% CI 1.37-18.33). Other characteristics of the IA were correlated with unruptured IA (lobulated wall aneurysm, PComA/AComA location and multiple aneurysms) and were founded to be independently associated with unruptured IA. Mortality was around 20% at one-year follow-up in ruptured IA. Authors concluded for a potential protective role of the association of statins and ASA treatment in IA rupture.

Although the work is interesting and substantially in line with recent literature findings, the paper is inaccurate in some parts and should be implemented in its formal presentation.

Q. Abstract- In conclusions section, authors stated that association treatment of statins plus ASA is more frequent in unruptured IA but this datum is not presented in the results section of the same abstract. It might be better to state that statins plus ASA association is independently associated with unruptured IA.

A. We have modified the conclusions section and introduce this idea in the abstract. Thanks.

Main text

Q. Background section should be more developed.

A.In this new version of the manuscript, we have included some new information in the Background section, following your recommendations.

Methods.

Q. How authors defined mortality? Did they consider mortality for all causes? Or related to SAH?

A.Mortality was predefined for all causes, although, in our study population, 100% (38) of patients died during the follow-up were related to the SAH.

Q. In the statistical analysis section, authors declared that risk factors were associated with IA rupture and Fisher 4 in ruptured IA, but the results of the association between IA rupture and Fisher 4 were not presented in text nor tables in the results section.

A.This sentence has been removed in this new version. In a previous version of the study, we found interesting to study the variables associated with the presence of Fisher 4, but finally, we decided to remove it from this paper, because our conclusions and interpretations could be ambiguous. We decided to focus on statins and ASA and write a different paper related to Fisher 4 and the potential variables associated to that. We are sorry for this confusing issue.

Q. Authors stated that a backward and forward stepwise procedure was used for multivariable logistic regression. Which approach was finally used in table 2?

A.We used the forward stepwise procedure. This data has been modified in the Statistical analysis section. Thanks.

Results.

Q. Authors stated that mortality is 19.6% in the ruptured IA at 12 months in the text and at 3 months in table 1. This discrepancy should be solved.

A.This data has been amended in this revised version of the manuscript. Thanks.

Q. Mean ASA dose should be reported.

A. The most frequent ASA dose was 100mg per day (94.7%) and Simvastatin was the statin most frequently used (52.4%). This data has been added in the revised manuscript version.

Q. Table 2. Statins treatment showed a significant different distribution in ruptured and unruptured IA (14.4 vs 26.2%, p=0.005) in univariate analysis (table 1). Why statins treatment alone was not inserted as independent variable in multivariable regression models?

A.We tried to compare combined treatment (ASA plus statins) with ASA alone, in order to evaluate the effect of the synergist treatment. However, we performed the analysis including the statins, but this variable was not included in the final Table 2. In this revised manuscript, we have included statins in this Table 2. Thanks. 

Discussion section should be more fluent and less structured to be more enjoyable for readers.

Disappointingly, findings were not really discussed according to literature. Here, some examples:

Q. Why ASA alone did not result protective in this population, as expected from literature data? Is it in relation with sample size? Is it related to selection bias? Why patients were taking ASA at the time of diagnosis? All these aspects should be discussed. Moreover, ASA dose should be reported and discussed since an inverse dose-response relationship with SAH risk has been reported (Reference 9).

A.In the univariate analysis, ASA was associated with unruptured IA state as previous studies, however, due to the high effect of the combined treatment (ASA plus statin) not evaluate so far, ASA as a single therapy, was not independently associated with IA rupture. We hypothesized that combined treatment is superior to any single therapy in preventing IA rupture, as a synergist effect. We have highlight this idea in the revised manuscript version.

Regarding the ASA dose, we include in this revised version that 36 of 38 patients were taking 100mg/day at the moment of the diagnosis, just 2 of the 38 were taking high high dose (300mg/day), so this asymmetrical distribution did not allow us to analyze this issue. According to Can et al (Reference 9) high ASA dose (325mg/day) could lead to an increasement in IA rupture, but with a high risk of rerupture. We take it into account this issue for our future investigations in this line and determine if we find differences in first IA rupture according to ASA dose. Thanks for this valuable comment.

The most frequents reason for taking ASA at the moment of the diagnosis was the presence of ischemic cardiopathy in 21 cases.

In this revised manuscript, we have discussed our findings and compare our results with previous studies. We have also modified the structure of the discussion.

Q. The same should be done for statins alone, on which far less evidence is available.

A.We have included statin in the multivariable analysis. Thanks.

Regarding the statins dose, we reported in our study population up to 8 differences doses and 5 different statins (Atorvastatin, Simvastatin, Pravastatin, Rosuvastatin and Fluvastatin). Unfortunately, the number of patients included do not allow us to perform an analyze by subgroups. Certainly, we detect a mild benefit of Atorvastatin in IA rupture compare to the others (23.3% vs 37.7%, p=0.135) but we cannot make a strong conclusion due to the small sample in this subgroup. 

Our primary goal was to evaluate the effect of a combined treatment (ASA plus statins). In the future, with more patients included in our database, we probably could evaluate if any specific statin or dose is associated with a smaller incidence of IA rupture.

Q. Another interesting point is that dome diameter of unruptured IA was significantly larger than ruptured IA, which is not expected. Moreover, the balanced distribution of high blood pressure between the two groups is not expected since well-controlled blood pressure are associated with a low risk of unruptured IA growth (Weng GC. Aspirin and Growth of Small Unruptured Intracranial Aneurysm: Results of a Prospective Cohort Study. Stroke. 2020 Oct;51(10):3045-3054. doi: 10.1161/STROKEAHA.120.029967. Epub 2020 Sep 3).

A. The importance of dome diameter and the risk of IA rupture is of our interest since some years. The study of new variables in predicting IA rupture such as ASA, statins, lobulated wall, aneurysm location, dome/parent artery ratio, angle inflow… have demonstrated to be superior in predicting IA rupture compare to dome diameter. In our experience, we see a lot of “small” aneurysms rupture but in dangerous locations like PCom and ACom, compare to those “big” aneurysm located at cavernous and paraophthalmic segments, that usually are detected due to III nerve palsy, or secondary to headaches, but without SAH condition. (Weir et al. J Neurosurg 96:64–70, 2002)

We think (and some other investigators have published similar data: Weir et al., Duan et al.) that location is probable more important than dome size, in order to better establish a risk rupture. Duan et al (Sci Rep 8, 6440 (2018)) reported no differences between dome size and IA rupture (4.21 vs 4.28, p=0.191), however size ratio (SR) and aneurysm inflow angle (FA) were identified as independent variables of IA rupture (OR: 3.586, 95% CI (1.518–8.474) and OR: 2.241, 95% CI (1.065–4.715))

In our series, HBP was not associated with IA rupture, in contrast to some previous studies, although, for some investigators, HBP could be associated with IA formation but not with IA rupture (Chalouhi et al. Stroke. 2013 Dec;44:3613-22.). Duan el at, neither identify HBP as an independent predictor of IA rupture (OR:1.505, 95% CI (0.849–2.668)). Weng et al, reported that uncontrolled hypertension is associated with IA growth, but antihypertensive consumption was not associated with IA growth, so we can conclude that uncontrolled hypertension, but not hypertension, is associated with IA growth (and probably rupture). Unfortunately, we did not collect the variable “uncontrolled hypertension” for this analysis. Interesting topic for our future investigations.

Thanks for this new reference, we have included both reference and discussion in the revised manuscript.

Q. Another main limitation that should be added is the cross- sectional nature of the study.

A.We have included it in the Limitation section.

Reviewer #2: The authors presents a single institution study of cerebral aneurysms and report that a combination of ASA and a statin is associated with a lower incidence of aneurysm rupture. Given the general interest in ASA and statins for preventative care of cerebral aneurysms, this manuscript is worthy of publication if the following comments are addressed.

1) A major limitation to this study is that the patient population is limited to patients who underwent treatment of their aneurysms. This data can therefore not be generalized to all cerebral aneurysm patients. Please discuss this in the "limitations and strengths" section.

A.Certainly, all the patients included in the study were treated endovascularly, so maybe some patients with IA who are not candidates for embolization, are not good represented in this study. We have included this issue in the revised manuscript.

2) Please expand your final Conclusion to provide a brief summary of your data (ie, that the combination of ASA and a statin was more frequently seen in unruptured aneurysms).

A.Thank you. We have modified the Conclusions section and we have included this part of our results.

3) Multiple aneurysms has been associated with a higher risk of aneurysm rupture, and yet you saw the opposite in your data. Do you have any thoughts on this?

A.This is an interesting topic that surely could lead to future papers. In our study population patients with multiple aneurysms were less frequent ASA users at the moment of the diagnosis (7.4% vs 11.1%, p=0.13) and with less AComA and PComA aneurysm location (30.5% vs 43.5%, p=0.005), so we consider that these two variables could influence in the results. We have included this observation in the Discussion section. Thanks.

4) Including Figure 1 places an emphasis on mortality data, although you found no statistically significant differences in mortality based on ASA and statin use. I would suggest either removing this figure and reporting this data in a table or text or adding error bars to the graph to make it more clear that there are no statistical differences.

A.Thanks for this recommendation. Following your instructions, we have removed this Figure and we have performed a new one in which we have included the rupture aneurysm rate differences according to the type of treatment at baseline. We find this Figure more illustrative and understandable than the previous one. Now is the new Figure 1 in the manuscript.

Reviewer #3: The authors present a well-written manuscript on the effects of statins and ASA treatment on the rupture of intracranial aneurysms. This is a retrospective review that looks at risk factors and treatment with either statin, ASA or a combination in two groups of aneurysms- ruptured and unruptured. Based on rupture status alone, the authors observe a protective effect of combination treatment. This is a modestly sized study that is retrospective in nature. Additionally there is no follow-up period for patients. I think looking at the data in terms of predictors (ie treatment with statin and/or ASA) does provide some valuable information, and provides some of the seed data needed to perform an RCT of these two medications for treatment. However, I think we still don't truly know from this study if the medications are protective because there is no way of know how long any of the patients harbored an aneurysm, and there is no data available to perform survival analysis with a Cox model. Nonetheless, I think this manuscript serves as an excellent review of the potential effects of ASA and statin on the biology of intracranial aneurysms, and the data helps provide more impetus to move towards an RCT. I would ask that the authors include some of my comments in the limitations section, and flesh out the limitations of the study further in the discussion. I think the conclusions are appropriate for the study design and data.

A.We appreciate so much your comments. Actually, we performed a follow-up of all patients at 6 and 12 months with at least 1 follow-up angiography in all survivors.

We completely agree, RCT are mandatory in this stage to better evaluate the role of these treatments in aneurysm rupture status. We have included your comments in the Limitations section in the revised manuscript and we have also modified the Discussion section. 

Thanks for your comments and inputs.

Our team is grateful about reviewers and editor comments, we think that the quality of the manuscript has improved and this new version is more understandable. Thanks so much.

---

## [Editor Report · Decision Letter 1]

3 Feb 2021

EFFECT OF COMBINED ACETYLSALICYLIC ACID AND STATINS TREATMENT ON INTRACRANIAL ANEURYSM RUPTURE

PONE-D-20-30637R1

Dear Dr. Silva,

We’re pleased to inform you that your manuscript has been judged scientifically suitable for publication and will be formally accepted for publication once it meets all outstanding technical requirements.

Kind regards,

Massimiliano Toscano

Academic Editor

PLOS ONE
---

## [Editor Report · Acceptance letter]

8 Feb 2021

PONE-D-20-30637R1 

Effect Of Combined Acetylsalicylic Acid And Statins Treatment On Intracranial Aneurysm Rupture. 

Dear Dr. Silva:

I'm pleased to inform you that your manuscript has been deemed suitable for publication in PLOS ONE. Congratulations! Your manuscript is now with our production department. 

Kind regards, 

on behalf of

Dr. Massimiliano Toscano 

Academic Editor

PLOS ONE